# MicroRNA and Protein Cargos of Human Limbal Epithelial Cell-Derived Exosomes and Their Regulatory Roles in Limbal Stromal Cells of Diabetic and Non-Diabetic Corneas

**DOI:** 10.3390/cells12212524

**Published:** 2023-10-25

**Authors:** Nagendra Verma, Drirh Khare, Adam J. Poe, Cynthia Amador, Sean Ghiam, Andrew Fealy, Shaghaiegh Ebrahimi, Odelia Shadrokh, Xue-Ying Song, Chintda Santiskulvong, Mitra Mastali, Sarah Parker, Aleksandr Stotland, Jennifer E. Van Eyk, Alexander V. Ljubimov, Mehrnoosh Saghizadeh

**Affiliations:** 1Eye Program, Board of Governors Regenerative Medicine Institute, Cedars Sinai Medical Center, 8700 Beverly Boulevard, AHSP-A8104, Los Angeles, CA 90048, USA; nagendra.verma@cshs.org (N.V.); drirhkhare@gmail.com (D.K.); cynthia.amador@cshs.org (C.A.); andrew.fealy23@gmail.com (A.F.); shaghaiegh.ebrahimi@cshs.org (S.E.); odeliashadrokh@gmail.com (O.S.); ljubimov@csmc.edu (A.V.L.); 2Departments of Biomedical Sciences, Cedars Sinai Medical Center, Los Angeles, CA 90048, USA; 3Division of Pediatric Blood and Marrow Transplantation & Cellular Therapy, University of Minnesota, Minneapolis, MN 55455, USA; 4Sackler School of Medicine, New York State/American Program of Tel Aviv University, Tel Aviv 6997801, Israel; 5Genomics Core, Cedars Sinai Medical Center, Los Angeles, CA 90048, USA; xueying.song@cshs.org (X.-Y.S.); chintda.santiskulvong@cshs.org (C.S.); 6Advanced Clinical Biosystems Research Institute, The Smidt Heart Institute, Cedars Sinai Medical Center, Los Angeles, CA 90048, USA; mitra.mastali@cshs.org (M.M.); sarah.parker@cshs.org (S.P.); aleksandr.stotland@cshs.org (A.S.); jennifer.vaneyk@cshs.org (J.E.V.E.); 7Department of Medicine, David Geffen School of Medicine at UCLA, Los Angeles, CA 90024, USA

**Keywords:** diabetic cornea, extracellular vesicles, exosome, cellular crosstalk, miRNA, limbal stem cells, mesenchymal stem cells, limbal epithelial cells, proteomics, RNA-seq

## Abstract

Epithelial and stromal/mesenchymal limbal stem cells contribute to corneal homeostasis and cell renewal. Extracellular vesicles (EVs), including exosomes (Exos), can be paracrine mediators of intercellular communication. Previously, we described cargos and regulatory roles of limbal stromal cell (LSC)-derived Exos in non-diabetic (N) and diabetic (DM) limbal epithelial cells (LECs). Presently, we quantify the miRNA and proteome profiles of human LEC-derived Exos and their regulatory roles in N- and DM-LSC. We revealed some miRNA and protein differences in DM vs. N-LEC-derived Exos’ cargos, including proteins involved in Exo biogenesis and packaging that may affect Exo production and ultimately cellular crosstalk and corneal function. Treatment by N-Exos, but not by DM-Exos, enhanced wound healing in cultured N-LSCs and increased proliferation rates in N and DM LSCs vs. corresponding untreated (control) cells. N-Exos-treated LSCs reduced the keratocyte markers ALDH3A1 and lumican and increased the MSC markers CD73, CD90, and CD105 vs. control LSCs. These being opposite to the changes quantified in wounded LSCs. Overall, N-LEC Exos have a more pronounced effect on LSC wound healing, proliferation, and stem cell marker expression than DM-LEC Exos. This suggests that regulatory miRNA and protein cargo differences in DM- vs. N-LEC-derived Exos could contribute to the disease state.

## 1. Introduction

The integrity of the corneal epithelium and stroma is essential for corneal transparency and visual function. Corneal epithelium is continuously regenerated by limbal epithelial stem cells (LESCs) located in the basal layer of limbus, a corneoscleral junctional zone [1,2]. LESCs have a lifetime capacity for self-renewal and the ability to generate transit-amplifying cells (TAC) that proliferate, migrate, and differentiate into central corneal epithelial cells [3,4] through their potential crosstalk with stromal cells [5,6,7]. Limbal niche is composed of both epithelial and stromal stem/progenitor cells and specialized extracellular matrix (ECM). They are involved in the homeostatic maintenance, proliferation, and migration of LESCs and multipotent limbal stromal stem cells (LSSCs). LSSCs express MSC markers (e.g., CD105, CD90, and CD73) and can differentiate into corneal keratocytes [8,9,10]. The hallmark feature of the limbal niche is the interaction or symbiotic relationship between LESCs and LSSCs/LSCs that are near stromal crypts housing LESCs [10]. The intercellular communication of LESCs and LSSCs/LSCs is well documented and occurs through direct cell–cell contact, secreted soluble factors, and recently emerged extracellular vesicles (EVs), which are vital for the stem cell maintenance and activation [5,6,7,11,12,13,14,15]. Any damage to LESCs or limbal stromal niche due to external insults or diseases such as diabetes may lead to a pathologically altered self-renewal and wound healing, and ultimately to compromised vision or even blindness [16,17,18,19]. Additionally, it may cause stromal remodeling as a result of epithelial damage and IL-1 release from epithelial cells or tears, resulting in apoptosis of stromal cells beneath the epithelial wounds [20,21,22]. During remodeling or wound healing, quiescent LSCs activate and transform to differentiated keratocytes. In turn, keratocytes differentiate to fibroblasts and myofibroblasts, potentially producing a stromal scar, which can lead to impaired transparency and vision [16,22,23]. Therefore, the health of LESCs and LSSCs/LSCs in their niche environments is a key factor in normal corneal homeostasis and wound healing. Emerging studies suggest that function of epithelial cells can be modulated by the underlying LSCs or vice versa through their secreted EVs without requiring direct contact between the donor and recipient cells [14,15,24,25]. 

EVs are heterogeneous vesicles (50–1000 nm), which differ by their subcellular origins, sizes, and modes of release [26,27]. EVs are released by all cell types and interact with recipient cells to mediate physiological intercellular crosstalk by transporting mRNA, DNA, microRNA, and protein cargos [28]. Exosomes (Exos), a type of nanosized EVs (50–150 nm), are generated inside endosomes, forming intraluminal vesicles within multivesicular bodies and released into the extracellular environment. This process involves different protein functional families such as endosomal sorting complex that is required for EV transport, Ras-related protein Rabs, heat shock proteins, tumor susceptibility gene 101 protein (TSG101), vacuolar protein sorting-associated proteins (VPSs), and lipids [29,30,31,32]. Exos contain a range of specific nucleic acids and proteins, which also can differ from cell and tissue of origin [28,33]. Packaging of RNA into Exos is facilitated by proteins such as annexins, major vault protein, lupus La protein, and heterogeneous nuclear ribonucleoproteins. They bind to the specific RNA sequence motifs or by recognizing unique secondary RNA structures and/or through specific RNA modifications [34,35,36,37]. Recent studies demonstrated that Exos’ RNA profiles are not a reflection of, but differ substantially from, their originating parental cells. This suggests the presence of regulatory mechanisms of cargo sorting of molecules into Exos [29,34]. EVs/Exos are involved in various physiological and pathological conditions such as cancer [38], cardiovascular diseases [39], diabetes [40,41,42], as well as in wound healing [43,44]. Understanding the mechanisms of EVs/Exos in limbal niche cell functioning in healthy, diseased, and healing corneas could be key to managing various aspects of corneal diseases.

Previously, we and others have shown that LSC-derived Exos function as a new form of intercellular communication required to maintain and regulate limbal epithelial cell (LEC) proliferation, migration, and differentiation in vitro and in ex vivo organ-cultured corneas [13,14,15,45,46]. Our study was the first to demonstrate the differential role of Exos in the limbal niche in LSC-LEC communications in diabetic vs. healthy corneas [15]. Reciprocal interaction between LECs and LSCs in the limbal niche is implicated in their communication and limbal homeostasis. Therefore, to acquire a comprehensive knowledge of their crosstalk, which is essential for our understanding of cornea in health and disease, we have now characterized non-diabetic (N) and diabetic (DM) human LEC-derived Exos using next generation sequencing (NGS) and proteomics (mass spectrometry (MS)) profiling. This allows for the identification of distinct miRNA and protein players and investigates their effects of diabetes on the LEC-derived Exos populations. Furthermore, we examined their roles in survival, migration, and proliferation of other cell types, LSCs, from non-diabetic and diabetic donors. Our study indicates that non-diabetic LEC-derived Exos (N-Exos) have a greater effect on cell proliferation and wound healing than diabetic LEC-derived Exos (DM-Exos). We have also documented differences in Exo cargos derived from non-diabetic and diabetic limbal epithelial cells.

## 2. Materials and Methods

### 2.1. Human Corneas Procurement

Age-matched human autopsy non-diabetic and diabetic corneas (Appendix A) were received from National Disease Research Interchange (NDRI, Philadelphia, PA, USA) in Optisol storage medium (Chiron Vision, Claremont, CA, USA) within 24–48 h of donor death (donor identity was withheld by the supplier). NDRI has a human tissue collection protocol approved by a managerial committee and subject to the oversight of the National Institutes of Health. In all cases, the required informed consent from donors’ next of kin specifying the use of postmortem tissue for research was obtained by NDRI-affiliated eye banks. The study reported here was conducted according to the guidelines of the Declaration of Helsinki and was covered by approved Cedars-Sinai Medical Center IRB protocol Pro00019393.

### 2.2. Isolation and Maintenance of Primary Human LECs and LSCs

Primary limbal epithelial and stromal cells were isolated from the age-matched autopsy non-diabetic and diabetic limbal rims, as described previously [15,47]. Briefly, LECs containing LESCs were removed from corneoscleral rims using Dispase II (2.4 U/mL, Roche Life Science, Pleasanton, CA, USA) for 2 h at 37 °C, followed by 0.25% trypsin–0.02% EDTA digestion for 30 min at room temperature. Cells were grown in EpiLife media containing Human Keratinocyte Growth Supplement (HKGS), N-2 supplement, B27 supplement, 1× antibiotic/antimycotic mixture, with added 15 ng/mL epidermal growth factor (EGF) in plates coated with a mixture of fibronectin, collagen IV, and limbal-expressed laminin-521 (Thermo Fisher Scientific, Waltham, MA, USA). Limbal stroma containing LSSC was chopped and incubated in 1 mg/mL collagenase type IV solution (STEMCELL Technologies, Seattle, WA, USA) at 37 °C overnight. LSCs were filtered, washed, and re-suspended in a complete culture medium (CCM) (DMEM/ F12 supplemented with B27, N2, 1% antibiotic/antimycotic, and 10 ng/mL basic fibroblast growth factor (FGF2), PeproTech Inc, Rocky Hills, NJ, USA), plated at 8 × 10^3^ cells/cm^2^ and kept in the incubator at 37 °C and 5% CO_2_. Both LECs and LSCs were used up to passage 3 at 70–80% confluence and passaged using TrypLE Express (Thermo Fisher Scientific).

### 2.3. Isolation and Purification of Exos from Primary LEC Culture Supernatants

Exosomes were isolated from conditioned media of N and DM LECs (passage 3) using the differential ultracentrifugation method. The conditioned media on each alternative day were collected 3–4 times until the cells reached about 90% confluency and subjected to successive centrifugation steps at 300× *g* for 10 min to eliminate cell debris and macro particles. Then, the supernatant was collected and centrifuged at 10,000× *g* for an additional 20 min. The resulting cell-free medium was ultra-centrifuged at 100,000× *g* for 1 h, and the collected exosome pellet was re-suspended in 1× PBS and ultra-centrifuged at 100,000× *g* for another hour. Finally, the Exo pellet was re-suspended in 1× PBS and pooled from the same flask and stored at −80 °C until further analysis.

### 2.4. Analysis of Exo Particle Size by Dynamic Light Scattering

Resuspended Exos were analyzed in real time using dynamic light scattering with Nano Sight LM10-HS instrument equipped with a laser (638 nm) and Nanoparticle Tracking Analysis software version 2.3, Build 0033 (NanoSight, Westborough, MA, USA). Post-acquisition settings were based on the manufacturer’s recommendations and kept constant between the samples. Each measurement was analyzed to obtain particle size distribution profiles and concentrations.

### 2.5. Flow Cytometry Analysis of Primary LSCs and Exo Characterization

Flow cytometry was performed as per the manufacturer’s instructions. Briefly, pooled Exos from LECs were incubated with 4 µL of 4 µm size aldehyde/sulfate latex beads 4% (*w*/*v*) (Thermo Fisher Scientific) for 15 min at room temperature (RT) and then overnight at 4 °C under mild agitation. Functional groups remaining on the beads were blocked by incubation with 100 mM glycine for 30 min at RT under mild agitation. Exosome-coated beads were incubated with PE-CD63 (H5C6) and APC-CD81(5A6) antibodies (Appendix A). Complexes were resuspended in isolation buffer and subjected to flow cytometry using BD LSR II instrument (BD Biosciences, Franklin Lakes, NJ, USA), where at least 50,000 events were collected, and results were analyzed by FlowJo software.

Cultured primary LSCs were harvested and fixed with fixation/permeabilization solution kit (BD Biosciences) and stained with PE-conjugated anti-human CD90 (BioLegend, San Diego, CA, USA), APC-conjugated anti-human CD105 (BD Pharmingen), and V450-conjugated anti-human CD73 antibodies (BD Horizon™) for 30 min at 4 °C in the dark. All antibodies were used at a final concentration of 5 µL/million cells. After staining, cells were washed with Perm/Wash Buffer (BD Biosciences) and analyzed by flow cytometry (BD LSR II instrument). Data were analyzed using FlowJo version 10.9 software (FlowJo LLC).

### 2.6. Western Blot Analysis

Western blot was performed as described previously [48,49]. Treated cells or isolated Exos were lysed in Tris-glycine sample buffer with protease inhibitor EDTA-free cocktail and boiled for 5 min at 100 °C. An equal quantity of lysates was loaded into 4–20% or 8–16% gradient Tris-glycine SDS polyacrylamide gels and transferred onto nitrocellulose membranes. Blots were blocked with 5% skimmed milk followed by incubation with primary antibodies (Appendix A) overnight at 4 °C. IRDye LiCor secondary antibodies (Li-Cor Biosciences, Lincoln, NE, USA) were used. Blots were imaged and quantified using the Odyssey CLX imaging system (Li-Cor Biosciences). The target protein band intensities were normalized by β-actin content.

### 2.7. Immunostaining

Cultured primary LSCs were fixed in 10% formalin for 5 min at room temperature, permeabilized in 0.5% Triton X-100 for 5 min at RT, and incubated with primary antibodies (Appendix A) in a blocking solution overnight at 4 °C, followed by 1 h incubation with secondary antibodies conjugated with either Alexa Fluor 488 or Alexa Fluor 594 (Abcam, Waltham, MA, USA) in the dark at RT. Slides were mounted with ProLong Gold Antifade Mounting with DAPI (Thermo Fisher Scientific). Negative controls without a primary antibody were included in each experiment.

### 2.8. EV Labeling and Cellular Uptake

To track LSC and LEC uptake of the LEC-derived Exos using confocal microcopy, Exos were labeled using PKH-67 green-fluorescent Cell Linker Kits (Sigma-Aldrich, St. Louis, MO, USA) according to the manufacturer’s instructions. Briefly, Exos were incubated with PKH-67 for 5 min at RT in the dark followed by two washes in PBS and centrifugation at 100,000× *g* for 1 h each time. Next, 25 μg/mL labeled Exos in a total volume of 100 μL were diluted in respective medium and added to N or DM LSC or LEC cultures for 6 h and washed prior to staining with 10 μM CellTrace Calcein red-orange AM (Thermo Fisher Scientific) at 37 °C for 30 min in the dark. Cells were briefly washed and examined under a Zeiss LSM-780 confocal microscope (BioSciences, Jena, Germany). The control group was cultured in the respective medium with an added 100 μL of PBS.

### 2.9. Small RNA Next Generation Sequencing (NGS)

Library Preparation and Sequencing. Small RNA sequencing was performed at the Genomics Core at Cedars-Sinai Medical Center. Total Exos RNA was isolated by miRCURY RNA Isolation Kit–Cell and Plant (Exiqon, Woburn, MA, USA) according to the manufacturer’s instructions. RNA integrity was analyzed on the 2100 Bioanalyzer using the RNA 6000 Pico Kit (Agilent Technologies, Santa Clara, CA, USA), and RNA was quantified using the Qubit RNA HS Assay Kit (ThermoFisher Scientific). The QIASeqTM miRNA Library Kit (Qiagen, Hilden, Germany) was used to prepare miRNA sequencing libraries. Final library concentrations were measured using the Qubit 1× dsDNA HS Assay kit. Library fragment size was determined on the 4200 TapeStation (Agilent Technologies). Libraries were sequenced on a NovaSeq 6000 (Illumina, San Diego, CA, USA) at an average sequencing depth of ~10 M reads/sample and 1 × 75 bp sequencing.

Data Analysis. The demultiplexed raw reads were uploaded to GeneGlobe Data Analysis Center (Qiagen) at https://www.qiagen.com/us/resources/geneglobe/ (accessed on 19 May 2021) for quality control, alignment, and expression quantification. Briefly, 3′ adapter and low-quality bases were trimmed off from reads first using cutadapt (version 1.13) with default settings [50], then reads with less than 16 bp insert sequences or with less than 10 bp UMI sequences were discarded. The remaining reads were collapsed to UMI counts and aligned to miRBase (release v21) mature and hairpin databases sequentially using Bowtie v1.2 [51]. The UMI counts of each miRNA molecule were counted, and the expressions of miRNAs were normalized based on total UMI counts for each sample. In addition, expression data were analyzed in conjunction with TargetScan 7.2 to increase the likelihood of finding direct miRNA targets. Data are available via public GEO repository under accession No. GSE24334.

### 2.10. Proteomics by Liquid Chromatography-(LC)-MS (LC-MS/MS) Analysis

Exos were isolated from individual non-diabetic (*n* = 5) and diabetic (*n* = 5) human primary LEC culture supernatants. Samples were processed using Protifi S-TRAP sample preparation and trypsin digestion workflow as follows: Sample was lysed in into 50 μL 6 M Urea/5% SDS lysis buffer, protein concentration was estimated using BCA assay (Pierce, Thermo Fisher Scientific), and 100 μg was aliquoted for digestion. Proteins were reduced with 100 mM DTT, alkylated with 200 mM IAA, and digested with 5 μg trypsin. Tryptic peptides were eluted from Protifi (Farmingdale, New York, NY, USA) columns, dried, and resuspended in 0.1% Formic Acid water at 1 μg/μL concentration prior to liquid chromatography-based mass spectrometry (LC-MS) analysis. Mass spectrometry data were acquired on Fusion Lumos Orbitrap (ThermoFisher) instrument. Desalted peptides were separated on an Ultimate 3000 liquid Chromatography system with a 60 min gradient. Peptides were separated on a preformed gradient (ranging from 0 to 45% organic phase) on a Pharmafluidics capLC column (ThermoFisher) at a flow rate of 9.5 μL/min. Source parameters included spray voltage at 3.9 kV and an ion transfer temp of 290 °C. MS1 resolution was set to 120,000, AGC was set to 600,000 (150% normalized AGC target) with maximum injection time of 50 ms, and RF lens % was 30. Mass range of 400–1000 and AGC target value for fragment spectra of 400% were used. Peptide ions were fragmented using HCD at a normalized collision energy of 30%. Fragmented ions were detected across 40 DIA windows of 15 Da. MS2 resolutions were set to 15,000 with a max injection time of 30 ms. All data were acquired in profile mode using positive polarity. A sample specific library was generated using DIA-Umpire [52], based signal extraction followed by matching of DIA-Umpire pseudospectra (from Q1 files only) using the Trans Proteomic Pipeline (TPP, v5.2.0), spectral matching algorithms Comet [53], and X!Tandem [54]. Peptide level target-decoy probability scoring was performed by peptide prophet in the TPP [55], run individually on each search algorithm run, and then results of multiple searches were combined using the TPP InterProphetParser 1.0 Peptides with probability >0.95 were compiled into a preliminary library using TPP SpectraST 5.0 and retention times were aligned to iRT using Biognosys iRT standard peptides (Biognosys, Schlieren, Switzerland). iRT-aligned libraries were consolidated and converted to TraML format, and randomized decoy sequences were appended. The sample specific library was then searched against each individual DIA file using openSWATH 1.0 peak picking and scoring algorithm [56]. Decoy–target probability modeling was performed using pyProphet algorithm [57], and results from individual files were aligned across the experiment using the TRIC workflow [58]. Following normalization to total MS2 signal, mapDIA [59] was used to perform protein abundance inference and statistical comparisons. Data are available via ProteomeXchange with identifier PXD040918 (https://www.proteomexchange.org/).

Pathway analysis was used to profile the molecular activities of the differentially expressed genes (DEG)/proteins cargos. Visualization and exploration of the global proteome and ClueGO pathway analysis were performed with Protein Interaction Network Extractor (PINE) software 2.3.1 [60].

### 2.11. In Vitro MTS Proliferation Assay

To assess the effect of LEC-derived Exos on LSC proliferation, MTS assay was performed as described previously [15]. Briefly, LSCs were seeded on 96-well plates at 5000/well, and 25 μg N/DM LEC-derived exosomes were added to the basal medium without growth factors. Proliferation was measured using CellTiter 96 Aqueous One Solution Cell Proliferation Assay (Promega, Madison, WI, USA). After 24 h, 20 μL of MTS reagent was gently added to 100 μL culture media and incubated in a cell culture incubator for 4 h. At the end of the 4 h incubation, the color change due to the reduction of formazan by live proliferating cells was detected with a microplate reader at 490 nM.

### 2.12. In Vitro Scratch Wound Assay

Confluent LSCs treated with 25 μg LEC-derived N/DM-Exos were scratch wounded using a 200 µL sterile pipette tip, washed, and photographed at time 0. The wounds were allowed to heal and photographed every 6 h. All images were then analyzed using ImageJ 1.53t software (NIH, Bethesda, MD, USA). The percent wound area was calculated with reference to time 0. All experiments were performed in triplicate [15].

### 2.13. Statistical Analysis

Experiments were analyzed by Student’s *t*-test for two groups or ANOVA for three or more groups, with *p* < 0.05 considered significant, using Prism6 (GraphPad Software, San Diego, CA, USA).

## 3. Results

### 3.1. Characterization of Primary Limbal Stromal Cells

Cultured corneal stromal cells isolated from the limbal region were analyzed by immunostaining and FACS. Immunocytochemistry showed positive staining for specific LSSC/MSC markers, CD73, CD90, CD105 [61,62], and for aldehyde dehydrogenase 3A1 (ALDH3A1) and lumican, which are specific for keratocytes [8,63,64] in both N and DM LSCs (Appendix A). The main difference in N and DM was the marked reduction in ALDH3A1 in diabetic cultures. Flow cytometry further confirmed the expression of CD73, CD90, and CD105 in LSCs (Appendix A).

### 3.2. Characterization of Non-Diabetic and Diabetic Human Primary LEC-Derived Exos

LEC-derived Exos from at least three biological replicates of each N and DM LECs were characterized using different analytical methods. EV sizes ranged between 50 and 200 nm by NanoSight technology (Figure 1A). There were no significant differences between N (ave. mean size, 151 nm) and DM (ave. mean size, 164) Exo sizes (*n* = 4, *p* > 0.05). Common Exo markers (CD63, CD81, and HSP70) were used to characterize non-diabetic and diabetic LEC-derived Exos using flow cytometry and Western blot. Flow cytometry showed that 99.4% of N and 99.7% of DM human LEC-derived EVs were positive for CD63, and 96.7% of N and 97.6% of DM human LEC-derived EVs were positive for CD81 (Figure 1B), with no significant difference between two groups (*n* = 3, *p* > 0.05) using BD LSR II instrument (BD Biosciences). Western analysis of EVs showed positive bands for Exo markers CD63 and heat shock protein HSP70 (Figure 1C). The data suggested that the majority of our isolated EVs were exosomes.

### 3.3. Exosome Uptake by Cultured LSCs

Previously, we have shown the internalization of LSC-derived Exos by human non-diabetic and DM LECs [15]. In the present study, to determine and confirm the internalization of LEC/LSC-derived Exos by human non-diabetic and DM limbal stromal cells, PKH-67 green-fluorescent dye was used to label Exos. Primary LSCs were incubated with 25 μg/mL PKH-labeled LEC- or LSC-derived Exos for 6 h. FACS analysis was used to demonstrate the uptake of Exos in live N (Figure 2A) and DM (Figure 2B) LSCs. There was more than five-fold higher internalization of LEC- than LSC-derived Exos in LSCs by FACS analysis. Further exosome uptake was examined by confocal microscopy (Zeiss LSM-780, BioSciences). LEC-derived Exo internalization was documented at 6 h post-incubation in N (Figure 2C) and DM (Figure 2D) LSCs.

### 3.4. Next-Generation Sequencing of Small RNA in Non-Diabetic and Diabetic LEC-Derived Exos

To identify differentially expressed small RNA, including miRNA in N and DM LEC-derived Exos, total RNA was isolated, sequenced, and analyzed using NGS. Principal component analysis (PCA) showed segregation of N and DM LEC-derived Exos (Figure 3A). The supervised analysis of NGS data identified a total of 2629 small RNAs, including miRNA, piRNA, and snoRNA in all samples with the average threshold of more than one unique molecule identifier (UMI) (Appendix A). Among these entities, 2432 were known miRNAs. A set of 90 (23 upregulated and 67 downregulated) miRNAs was identified as differentially expressed in DM-Exos vs. N-Exos with the false discovery rate (FDR)-adjusted *p* < 0.1 and fold change of greater or less than 1.5 (Figure 3B, Appendix A). The volcano plot shows visual identification of miRNAs with large magnitude changes, which are also statistically significant (Figure 3C). The plot is constructed by plotting the *p* value (−log10) on the Y axis and the expression fold change between the two experimental groups (DM-Exos vs. N-Exos) on the X axis. The top of the plot (high statistical significance) and the extreme left or right (strongly down- and up-regulated, respectively) are the two regions of interest (Figure 3C). The top scoring differentially expressed miRNAs, which are all downregulated in DM LEC-derived Exos, such as miR-381-3p, miR-199a/b, miR-134-5p, miR-152-3p, and miR-34a-5p (Table 1), are involved in regulating cell cycles, TGF-β signaling, FGF, DNA methylation, PI3K/AKT, Notch signaling, and wound healing. There have been many studies, including ours [16,20], that have shown the important role and altered expression of these pathways in the diabetic cornea. To investigate if the Exos’ miRNA cargos represent the original cell content, we compared the top differentially expressed miRNAs in DM- vs. N-Exos with their corresponding miRNA expression levels in the DM- vs. N-LECs using our previous small RNA-seq data of N and DM human corneal limbus [47]. Interestingly, there was no correlations between their differentially expressed levels in Exos’ cargos and the originating LEC content (Table 2). Notably, all the 10 top significantly differentially expressed miRNAs were downregulated in DM- vs. N-Exos. Additionally, our small RNA-seq data indicated that 69% (*p* < 0.05) and 77% (adj *p* < 0.1) of differentially expressed miRNAs were downregulated in DM- vs. N-Exos.

Pathway Analysis: Ingenuity pathway analysis was performed to identify GO terms that are significantly associated with differentially expressed miRNAs in DM- vs. N-Exos identified to their target genes. Using miRSearch, we mapped the differentially expressed miRNAs (Appendix A) to their target genes and investigated whether specific GO terms were associated with these miRNAs (Appendix A). Figure 3D shows the top 20 selected pathways, such as ID1 (Inhibitor of Differentiation 1), PTEN signaling, autophagy, cell cycle, FGF, TGF, VEGF, and wound healing signaling pathways for the target genes found to be differentially expressed in DM- vs. N-Exos. Interestingly, all these pathways play important roles in self-renewal capacity, limbal epithelial, stromal homeostasis, and wound healing, which are altered in the diabetic cornea.

### 3.5. Proteomics Analysis of Non-Diabetic and Diabetic LEC-Derived Exo Cargos

Normal and diabetic primary LEC-derived Exos were analyzed for differential expression of proteins by LC-MS/MS. A total of 2648 unambiguous proteins were identified associated with Exos in all samples (Appendix A). A set of 314 (60 upregulated and 254 downregulated) proteins was differentially expressed in DM- vs. N-Exos with a raw *p* < 0.05 and fold change (FC) of ±1.5 (Appendix A). Of these 314 proteins, 34 proteins (5 upregulated and 29 downregulated) showed a false discovery rate (FDR)-adjusted *p* < 0.1. A heatmap of 50 differentially expressed proteomic targets in DM- vs. N-Exo divided two groups into distinct clusters (Figure 4A). The volcano plot shows visual identification of proteomic targets with large-magnitude changes, which are also statistically significant (Figure 4B). Table 3 shows the top scoring differentially expressed proteomic targets in DM vs. N-LEC-derived Exos. Strikingly, most of the top-scoring differentially expressed proteins (Table 3), which are mostly downregulated in DM-Exos, such as NEDD8-activating enzyme E1 regulatory subunit (ULA1), heat shock proteins (HSP 90A, 90B), ribosomal proteins (RL24, RSS9), 14-3-3 protein theta (1433T), and upregulated annexin A4 (ANXA4), have been implicated in the regulation of exosome production and cargo sorting [29,37,65]. Furthermore, we have shown in Table 4 the list of different categories of selectively enriched proteins in Exos, which are significantly differentially expressed (*p* < 0.05) in DM vs. N LEC-derived Exos. Noteworthy, 75–85% of the differentially expressed proteins (*p* < 0.05 and adj *p* < 0.1, respectively) were downregulated in DM- vs. N-Exos.

Pathway analysis: PINE pathway analysis of differentially expressed proteins in DM-Exos vs. N-Exos (Figure 4C, Appendix A) showed significant differences in VEGF and MTOR signaling, apoptosis, translation, metabolism, and cellular responses to stress pathways, which all play important roles in corneal epithelial and stromal homeostasis.

### 3.6. Non-Diabetic Human LEC-Derived Exos Induced Wound Closure and Proliferation in Primary LSCs In Vitro

N-LSCs or DM-LSCs treated with 25 μg/mL N-Exos significantly increased proliferation rate compared to their corresponding untreated control cells (Figure 5A). However, there were no significant changes in proliferation rate in DM-Exos treated N-LSCs or DM-LSCs compared to their corresponding untreated control cells (Figure 5A). Cell migration and wound closure were significantly enhanced in primary N-LSCs treated with N-Exos compared to untreated control cells, whereas DM-Exos treatment did not change the wound healing rate compared to control (Figure 5B).

### 3.7. Effects of Non-Diabetic and Diabetic Exos’ Cargos on Activation of Wound Healing-Related Signaling Molecules in Primary LSC

Mitogen-activated protein kinase 1 (MAPK1/ERK) was among the top scoring differentially expressed proteomic targets and was downregulated in DM- vs. N-LEC-derived Exos (Table 3). Western analysis confirmed and validated the downregulation of ERK1/2 in DM- vs. N-LEC-derived Exos’ cargos and the upregulation of ALDH3A1 expression level (Figure 6A). To examine the effect of N-Exos and DM-Exos cargos on signaling pathways during wound healing, wounded N-LSC cultures were treated with either N- or DM-Exos. Western analysis of wound healing related signaling molecules showed significantly increased expression of phosphorylated/activated p-ERK1/2 in wounded LSCs incubated with N-Exos compared to DM-Exos or control untreated cells, whereas DM-Exos treatment did not change the p-ERK1/2 expression compared to control untreated cells (Figure 6B). There was a moderate increase but no significant difference in levels of p-Akt in N- or DM-Exos treated vs. untreated wounded LSC cultures (Figure 6C).

### 3.8. Effects of N and DM LEC-Derived Exos on the Expression of MSC and Keratocyte Markers of Cultured Non-Wounded and Wounded LSCs

The effects of N and DM LEC-derived Exos on the expression of putative MSC and keratocyte markers were examined in primary non-wounded and wounded LSC cultures by Western blot. The changes in protein levels of keratocyte or MSC markers did not reach significance in both wounded and non-wounded LSC cultures treated with N- or DM-Exos vs. non-treated control (Figure 7). However, N-Exo treated non-wounded LSCs consistently showed decreased protein levels of keratocyte markers ALDH3A and lumican and modestly increased level of MSC markers, CD73, CD 90, and CD105 vs. control untreated cells (Figure 7A,B). On the contrary, wounded LSCs treated with N-Exos showed modestly increased protein levels of lumican and keratocan and decreased protein levels of CD90 and CD105 vs. untreated control wounded cells (Figure 7C,D).

## 4. Discussion

EVs play a major role in cellular homeostasis by their formation and release into extracellular environment leading to their uptake by neighboring and/or distant cells to balance intra- and extracellular signals. Exosomes, a subpopulation of EVs, contain complex molecular components, which include general and cell type-specific lipids, proteins, mRNA, and miRNA, enabling them to function as vectorized, multi-signaling devices [66,67,68]. Therefore, their regulatory roles make them likely contributors of physiological and pathological states. We previously reported for the first time Exos’ role in the limbal niche in LSC-LEC communications in healthy and diabetic corneas. We have shown that DM-LSC-derived Exos have distinct cargo profiles and differentially expressed small RNAs, including miRNA, compared to N-LSC-derived Exos. We have demonstrated exogenous Dil-labeled Exos uptake by immortalized human corneal epithelial cells, primary LECs in vitro, and limbal regions of ex vivo human organ-cultured corneas [15]. Additionally, our previous study showed that when LECs were treated with DM-LSC-derived Exos, the LECs showed reduced migration and proliferation, as well as altered marker expression, when compared to LECs treated with N LSC-derived Exos in vitro and in ex vivo organ-cultured corneas. Furthermore, we normalized wound healing in diabetic LECs by treating them with N LSC-derived Exos [15]. This observation suggested that the DM LSC-derived Exos could play a role in the pathogenesis of diabetic keratopathy.

In the present study, we continue to investigate and corroborate our previous findings of limbal cellular crosstalk in health and disease. We sought to elucidate reciprocal cellular communication and the effects of both N and DM LEC-derived Exos’ cargos on the phenotype of the other cell type (LSC), including migration, proliferation, wound healing, LSSCs/MSCs, and keratocyte marker expressions. This reciprocal interaction between LECs and LSCs in the limbal niche is implicated in cellular crosstalk and limbal homeostasis. We isolated and characterized the EV subpopulation derived from both non-diabetic and diabetic human LECs. They were primarily exosomes and were positive for documented exosome markers (CD63, CD81, and HSP70) and were within the expected size range (50–200 nm). To examine LSC-LEC crosstalk by Exos, we showed the uptake of exogenous PKH-labeled LEC- or LSC-derived Exos by primary LSCs. The five-fold higher internalization of LEC- than of LSC-derived Exos by N/DM primary LSCs (paracrine vs. autocrine, respectively) further confirmed the crosstalk between limbal epithelial and stromal cells (Figure 2).

Large amounts of miRNA cargos in EVs, including Exos, suggest their potential regulatory effects in recipient cell gene expressions [69,70]. In addition, the protein cargos in Exos may also exert various effects in recipient cells. To identify N- and DM-LEC-derived Exos’ cargos, we characterized their miRNA and proteins using small RNA sequencing and LC-MS/MS, respectively. We identified 90 (adj-*p* < 0.1) miRNAs and 34 (adj-*p* < 0.1) proteins as differentially expressed in LEC-derived DM-Exo vs. N-Exos. Among the top significantly differentially expressed miRNAs, a set of identified miRNAs downregulated in DM-Exo belongs to a highly conserved miRNA family, miR-199, consisting of miR-199a and miR-199b. Their major roles in the regulation of normal cell homeostasis in physiological and pathological processes, promoting cell proliferation, migration, and regeneration, have been documented [71,72,73,74]. Studies have shown the critical role of epigenetics, including DNA methylation, in the pathogenesis of diabetic state [19,20,75,76]. The fact that miR-152-3p expression, which regulates DNA methylation by targeting DNMT1 [77,78], is altered in DM-LEC-derived Exos, further suggests that the difference in DM- vs. N-Exos’ cargos could contribute to the diabetic disease state. Several studies, including ours, have shown the important role of Notch signaling, which is targeted by differentially expressed miR-34a-5p in DM-LEC-derived Exos in limbal niche homeostasis, including stem cell maintenance and differentiation [79,80,81,82]. These data suggest the crosstalk and association among the limbal cells and possible role of miR-34a-5p in LEC/LSC/MSC homeostasis and function. Furthermore, the most significantly altered pathways of differentially expressed miRNA target genes (Figure 3C) include cell cycle regulators, signaling molecules, wound healing, and inhibitors of the DNA binding 1 (ID1) pathway, which is mainly involved in cell cycling, migration, and differentiation.

Interestingly, small RNA-seq analysis of N and DM LEC-derived Exos and their cells of origin, LECs, showed no correlation between their miRNA expression levels (Table 2). Our data are in line with the growing evidence suggesting that cargo packaging is not a random process or replication of their original cell content and that there is an active cargo sorting machinery that controls the components of Exo generation [29,42]. Therefore, the miRNA profile differences in LEC-derived Exos from their cells of origin suggest that miRNAs are selectively incorporated into LEC-derived Exos [29,34,37,42]. In addition, the regulatory roles of differentially expressed proteins in different categories involved in Exos biogenesis and cargo sorting in DM- vs. N-Exos (Table 3 and Table 4) further support our genomic data and existence of mechanism that controls the sorting of bioactive molecules into Exos in LECs. Altered expression and/or dysregulation of these exosomal proteins that represent the originating cells, such as vacuolar protein sorting-associated proteins (VPSs), Ras-related protein Rabs (RABs), heat shock proteins (HSPs), and annexins (ANXAs), could lead to altered sorting of RNA and protein into Exos. The altered expression of proteins involved in biogenesis and cargo sorting machinery of Exos may also explain the higher percentage of downregulated miRNAs and proteins in DM- vs. N-Exos.

Some other significantly abnormally expressed proteins in DM- vs. N-LEC-derived Exos (Table 3) are mainly part of the signaling pathways involved in cell cycle, proliferation, migration, and survival, which are all altered in the diabetic cornea, resulting in delayed wound healing [20]. One of the significantly downregulated proteins is mitogen-activated protein kinase 1 (MAPK 1), as it has been also shown among top scoring differential target genes of miR-381-3p (Table 3). MAPK/ERK may emerge as one of the major cell type-specific altered pathways in LEC–LSC intercellular communications in diabetic corneas, which has been shown by p-ERK1/2 upregulation in N- vs. DM-Exos, as well as in their targeted treated cells, LSCs (Figure 6).

The presented data suggest that dysregulation of Exos biogenesis, packaging, and cargo sorting may contribute to their pathogenic roles under diabetic conditions. Although Exos contain specific contents of proteins and RNA that can be different from the originating LECs, the intraluminal content of the exosomal membrane depends on the originating cells, which is also affected by diabetes [42,83]. Ultimately, the exosomal cargos are an important determining element in their function that could reprogram recipient cells. Exosomal miRNAs may be the mediators that make the connection between the cells in the limbal niche and their downregulation in DM-Exos may lead to impaired homeostasis promoting diabetic disease features such as altered proliferation and migration. Further investigation is needed to elucidate the Exo biogenesis, packaging, and cargo sorting, as well as the function of their differentially expressed miRNAs and proteins in DM- vs. N-Exos in the limbal niche.

To validate the regulatory role of LEC-derived Exos’ cargos on their recipient LSCs, we performed a set of functional analyses. Importantly, healing was significantly promoted in wounded primary LSCs when incubated with N-LEC-derived Exos compared to control (untreated) wounded cells. However, DM-LEC-derived Exos did not significantly stimulate healing in wounded LSCs vs. control. The proliferation rate was also significantly increased in N- and DM-Exos treated primary N-LSCs compared to untreated cells, whereas there were no significant changes in the DM-LSC proliferation rate treated with DM-LEC-derived Exos compared to untreated cells. Thus, N-Exos have greater effects in stimulating wound healing and cell proliferation than DM-Exos in N- and/or DM-LSCs. These data confirm our previous study [15], in addition to similar studies in other cell types [43,64,84] and corneas [14,85], that Exos have regulatory roles in cell repair and wound healing through their cargo effects in the recipient cells. The difference in DM- vs. N-Exos’ effects is probably due to the differences in their cargo such as MAPK/ERK transferred from the donor cells to the recipient cells. It may be suggested that the difference in Exos’ cargos derived from N- and DM-LECs contributes to the disease state. Further studies are in progress to elucidate the role of other signaling molecules and/or specific miRNAs regulating signaling pathways, which may also contribute to stimulating wound healing or proliferation in their wounded target cells.

## 5. Conclusions

This study is the first to examine LEC–LSC crosstalk in the human limbal niche via LEC-derived Exos in non-diabetic and diabetic corneal cells. It is also the first study describing the small RNA transcriptome and proteome profile differences in N- and DM-LEC-derived Exos and their regulatory roles in LSCs in the limbal niche. Previously, we have reported cellular communication between LECs and LSCs in the limbus via LSC-derived Exos. In the present study, we revealed their interaction via LEC-derived Exos showing the bidirectional interaction of these progenitor cells.

We have also demonstrated the small RNA profile differences in LEC-derived Exos from their cells of origin for the first time, suggesting that miRNAs may be selectively incorporated into Exos. We have revealed the differential DM vs. N Exos’ influence on migration, proliferation, and LSC/MSC marker expressions in vitro. Furthermore, we identified differentially expressed protein and miRNA cargos in DM- vs. N-Exo derived from LEC, which may have roles in disease state. Our study was limited to primary cell cultures. The employed strategy and promising data on the differences between N- and DM-Exos are now needed to be confirmed in corneal organ cultures with natural limbal structure. Further investigation is required to elucidate the mechanisms of action of differentially expressed miRNAs and protein cargos in the heterogenous state of the diabetic limbal niche. The presented data provide new insights into intercellular communication by EVs and may help us to develop novel exosome-based tissue specific therapies in corneal pathologies.

## Figures and Tables

**Figure 1 cells-12-02524-f001:**
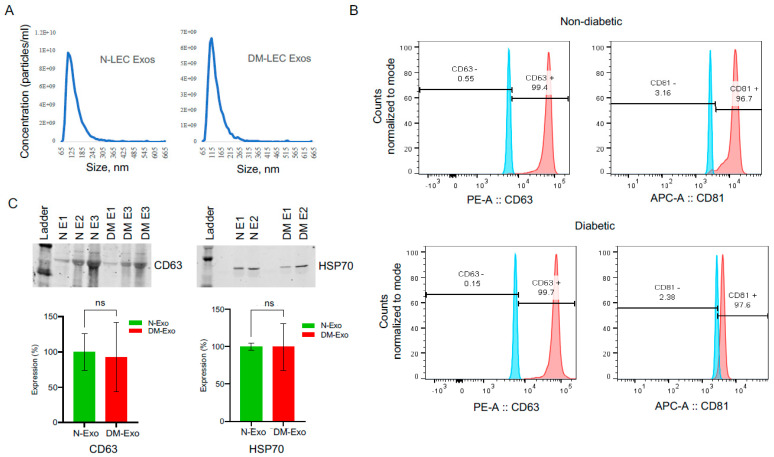
LEC-derived exosome characterization. (**A**) Size distribution of LEC-Exos was measured by NanoSight LM10. Histogram showed particle size distribution typical of exosomes. (**B**) Flow cytometry was performed on non-diabetic (N) and diabetic (DM) EVs using aldehyde sulphate latex beads. Vesicles were immunostained for CD63 (red) and CD81 (red) and compared with appropriate isotype control (blue), *n* = 3. (**C**) Western blot shows the expression of typical exosome markers CD63 and HSP70 in both N and DM vesicles. There were no significant changes in CD63 or HSP70 expression levels in DM-Exos compared to their corresponding N-Exos, *p* values 0.90 and 0.99, respectively. The bar graph represents the average SEM of pooled values of three independent triplicate assays, using a one-way ANOVA test. *n* = 3. N, non-diabetic; DM, diabetic; and E, Exosomes.

**Figure 2 cells-12-02524-f002:**
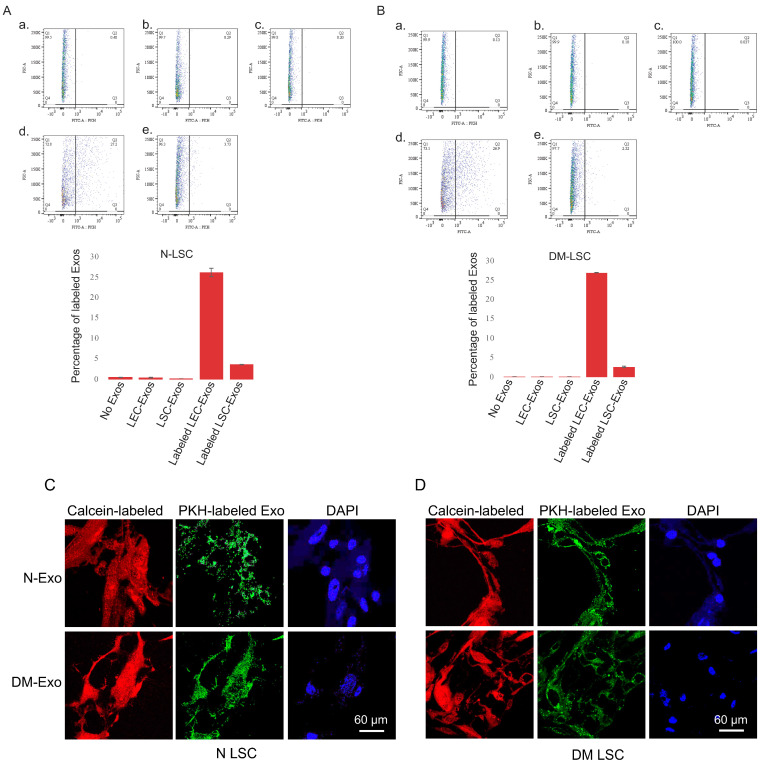
LEC- and LSC-derived EVs/Exos internalization. FACS analysis of PKH-labeled non-diabetic (N) LEC- and LSC-derived exosomes uptake by (**A**) N and (**B**) DM LSC. a. No Exo, control; b. unlabeled LEC-derived Exo; c. unlabeled LSC-derived Exo; d. labeled (Lab.) LEC-derived Exo; and e. labeled LSC-derived Exo. All experiments (*n* = 3) were performed in triplicate. FACS analysis showed more than five-fold increase in LEC- than LSC-derived Exos internalization in primary limbal stomal cells. (**C**) Immunostaining of LEC-derived exosome (Exo) uptake by non-diabetic (N) and diabetic (DM) LSCs. Confocal image of N- or DM-LSCs treated with 25 μg/mL PKH-labeled N or DM LEC-derived Exos for 6 h. PKH-labeled N- and DM LEC-derived Exos can be internalized by human primary LSC. The cells were stained with Calcein-AM (red fluorescence) that demonstrates live cells and their uptake of PKH-labeled Exos (green). Nuclei are stained with DAPI (blue).

**Figure 3 cells-12-02524-f003:**
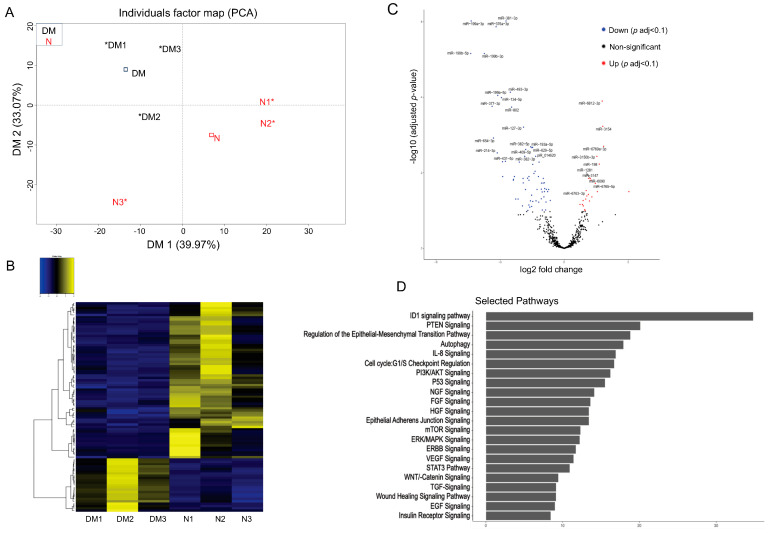
Exosome genomics analysis: Differentially expressed miRNAs in Diabetic (DM) vs. Non-diabetic (N) limbal epithelial cell (LEC)-derived Exo cargos. (**A**) Principal component analysis (PCA) based on the top 500 genes by variance across all samples, DM (Black) and N (Red), * individual cases of DM or N. (**B**) A total of 90 miRNAs were identified as significant DEGs (where 23 were upregulated, and 67 were downregulated). Two-way hierarchical clustering plots were utilized to make the Heat Map. (**C**) Volcano plot representing the differentially expressed miRNAs in DM vs. N LEC-derived Exos. The Y-axis corresponds to the mean expression value of log 10 (*p*-value), and the X-axis displays the log2 fold change. Genes that are identified as significant are colored in Red (upregulated) and Blue (downregulated). (**D**) Pathway analysis of differentially expressed miRNA targets in DM vs. N LEC-derived Exos showed significant pathways that are involved in corneal pathophysiology and homeostasis.

**Figure 4 cells-12-02524-f004:**
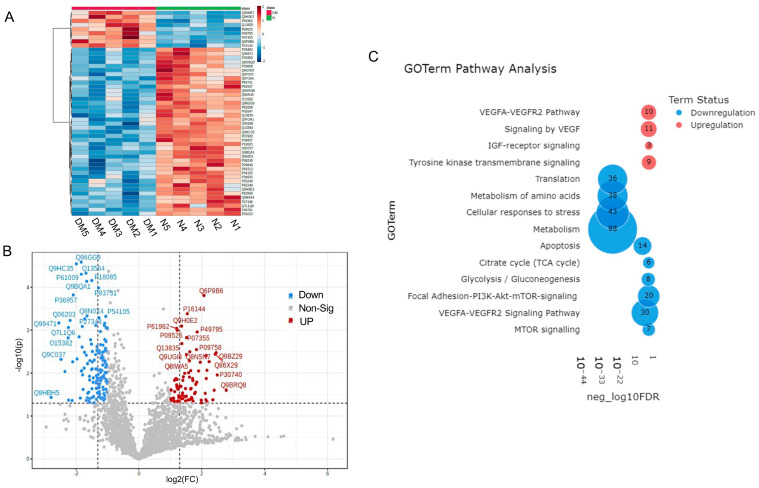
Proteomics analysis of differentially expressed protein cargos in DM vs. N LEC-derivExos. (**A**) Two-way hierarchical clustering plots (Heatmap). (**B**) Volcano plot of differentially expressed proteins. (**C**) PINE pathway analysis showed significant pathways involved in corneal pathophysiology and homeostasis.

**Figure 5 cells-12-02524-f005:**
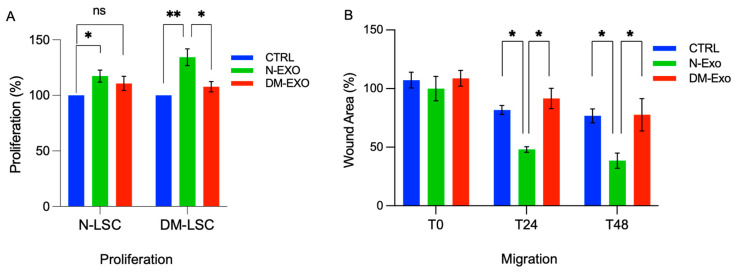
Non-diabetic LEC-derived EVs/Exos increase migration and proliferation of primary normal LSCs. (**A**) By MTS assay, proliferation rate is significantly increased in both normal (N-LSC) and diabetic (DM-LSC) LSCs treated with 25 μg/mL N LEC-derived Exos (N-Exo) compared to their corresponding untreated control cells. However, there were no significant changes in proliferation rate in N-LSCs or DM-LSC treated with DM LEC-derived Exos (DM-Exo) compared to their corresponding untreated control cells. The bar graph represents the average SEM of pooled values of three independent triplicate assays, compared to untreated cells (negative control), using a one-way ANOVA test. (**B**) Primary N-LSCs were scratch wounded and incubated with 25 μg/mL N- or DM-Exos, and wound closure was quantified using ImageJ software at 24 and 48 h after wounding. Cell migration and wound closure were significantly enhanced in N-LSCs treated with N-Exos compared to the cells treated with DM-Exos or PBS/untreated control cells. DM-Exos treatments did not change the wound healing rate compared to control. The bar graph represents average ± SEM of pooled values of three independent triplicate assays and compared to untreated control cells (negative control) by paired two-tailed *t*-test. ns (non-significant), * *p* < 0.05, ** *p* < 0.01.

**Figure 6 cells-12-02524-f006:**
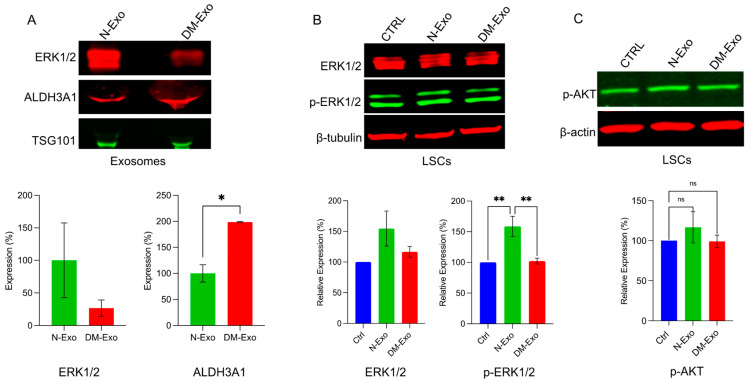
Western blot analysis of p-ERK1/2 expression in LEC-derived Exos and wounded LSC treated with normal or DM LEC-derived Exos. Total extracted protein from LEC-derived Exos (**A**) and wounded primary LSCs treated with N/DM Exos and control untreated cells (**B**,**C**) was separated on gradient SDS-PAGE gels, transferred to nitrocellulose and probed with selected antibodies. (**A**) Western blot showed increased expression level of ERK1/2 and significantly decreased expression of ALDH3A1 (as a control positive upregulated Exos’ cargo) in N-Exos vs. DM-Exos. TSG was used as a positive marker for Exos. (**B**) N-Exos treatment significantly increased protein levels of p-ERK1/2 vs. control (untreated) and DM-Exos treated LSCs. (**C**) There were no significant differences in levels of p-Akt in N- or DM-Exo treated vs. untreated wounded LSC cultures. Antibodies to β-actin or β-tubulin were used as equal loading controls and for semi-quantitation. All experiments were performed in triplicate. * *p* < 0.05, ** *p* < 0.01 by paired two-tailed *t* test.

**Figure 7 cells-12-02524-f007:**
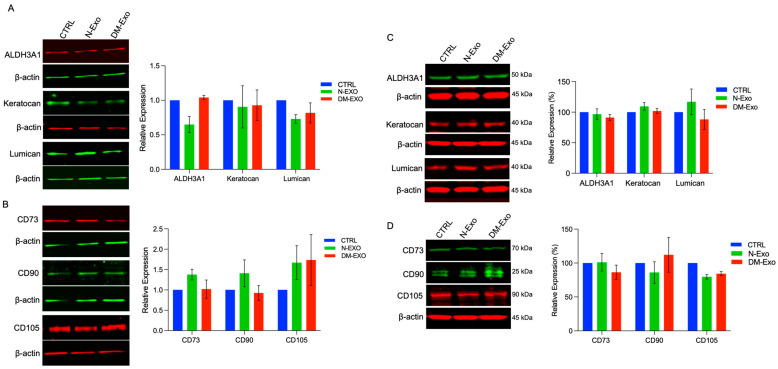
Western blot analysis of MSC and keratocyte markers in non-wounded and wounded LSCs treated with non-diabetic (N) or diabetic (DM) Exos. Total extracted protein from non-wounded (**A**,**B**) and wounded LSCs (**C**,**D**) treated with N- or DM-Exos and untreated cells (control, CTRL) was separated on gradient SDS-PAGE gels, transferred to nitrocellulose, and probed with selected antibodies. (**A**) N-Exos treatment modestly decreases protein levels of differentiated keratocyte markers, ALDH3A1 and lumican, but not keratocan vs. control and DM-Exos treated cells. (**B**) N-Exos treatment modestly increases protein levels of MSC markers, CD73, CD90, and CD105, vs. control. In contrast, DM-Exo treatment decreases CD73 and CD90 but increases CD105 protein expression levels. (**C**) N-Exos treatment (wounded LSCs) modestly increases protein levels of differentiated keratocyte markers, ALDH3A1 and keratocan. (**D**) N-Exos treatment (wounded LSCs) modestly decreases protein levels of MSC markers, CD90 and CD105, vs. control. However, none of the differences are significant. Antibodies to β-actin were used as equal loading control and for semi-quantitation. All experiments were performed in triplicate. Paired two-tailed *t* test. ns (non-significant).

**Table 1 cells-12-02524-t001:** Top significant differentially expressed miRNAs in DM- vs. N LEC-derived Exos and their fold changes (FC), *p* values, and targeted pathways.

miRNA	Log2(FC)	FC	*p* Value	*p* adj	Targeted Pathways
miR-381-3p	−2.725	0.151	1.47 × 10^−9^	9.51 × 10^−7^	TGF-β signaling, FGFR2/ p-MEK/p-ERK1/2
miR-199a-3p	−4.354	0.049	2.76 × 10^−9^	9.51 × 10^−7^	mTOR, CNTF & EGF signaling, Insulin receptor signaling, NGF/TrkA signaling, TGF-β pathway, PI3K/AKT/NF-κB
miR-199b-3p	−3.739	0.075	5.02 × 10^−8^	6.9 × 10^−6^	Wnt/β-catenin signaling, Nrf2 pathway, MAPK/ERK/EGR1
miR-199b-5p	−4.377	0.048	4.39 × 10^−8^	6.92 × 10^−6^	ERK/MAPK Signaling, FAK signaling, HGF signaling, p53 signaling, CDK5 signaling, Wound healing signaling pathway
miR-493-3p	−2.523	0.174	6.41 × 10^−7^	7.36 × 10^−5^	STAT1-TINCR-USP20-PD-L1, Akt/eNOS, CDKN1B/NF-κB, AKT1/mTORC1 signaling
miR-199a-5p	−3.111	0.116	9.06 × 10^−7^	8.91 × 10^−5^	STAT1-TINCR-USP20-PD-L1, Akt/eNOS, CDKN1B/NF-κB, AKT1/mTORC1 signaling
miR-134-5p	−2.937	0.131	1.19 × 10^−6^	0.0001	PI3K/AKT Signaling, Role of NANOG and OCT4 in embryonic stem cell pluripotency, WNT/β-catenin signaling
miR-218-5p	−2.883	0.136	0.0001	0.005	Cell Cycle: G1/S checkpoint regulation, Role of OCT4 in mammalian embryonic stem cell pluripotency
miR-152-3p	−1.175	0.443	0.0002	0.005	DNA methylation and transcriptional repression signaling
miR-34a-5p	−1.361	0.389	0.0004	0.009	Notch signaling

**Table 2 cells-12-02524-t002:** Comparison of top significantly differentially expressed miRNAs in DM vs. N-Exo with their corresponding differentially expressed miRNA in LEC. Fold change (FC), NA, not available.

miRNA	DM-Exo vs. N-Exo			DM-LEC vs. N-LEC		
	FC	*p* Value	*p* adj	FC	*p* Value	*p* adj
miR-381-3p	0.151	1.47 × 10^−9^	9.51 × 10^−7^	1.791	0.0139	0.9905
miR-199a-3p	0.049	2.76 × 10^−9^	9.51 × 10^−7^	NA	NA	NA
miR-199b-3p	0.075	5.02 × 10^−8^	6.9 × 10^−6^	1.110	0.3592	0.9905
miR-199b-5p	0.048	4.39 × 10^−8^	6.92 × 10^−6^	1.029	0.8650	0.9905
miR-493-3p	0.174	6.41 × 10^−7^	7.36 × 10^−5^	NA	NA	NA
miR-199a-5p	0.116	9.06 × 10^−7^	8.91 × 10^−5^	1.025	0.8029	0.9905
miR-134-5p	0.131	1.19 × 10^−6^	0.0001	2.371	0.0013	0.7816
miR-218-5p	0.136	0.0001	0.005	1.276	0.3765	0.9905
miR-152-3p	0.443	0.0002	0.005	NA	NA	NA
miR-34a-5p	0.389	0.0004	0.009	0.995	0.9816	0.9970

**Table 3 cells-12-02524-t003:** Top-scoring differentially expressed proteins in DM vs. N LEC-derived Exos with their gene and protein names, fold changes (FC), and *p* values.

Uniprot ID	Gene	Protein Name	log2(FC)	FC	*p* Value	FDR
Q9HC35	*EMAL4*	Echinoderm microtubule-associated protein-like 4	−2.469	0.181	0.00002	0.0263
P61009	*SPCS3*	Signal peptidase complex subunit 3	−2.587	0.166	0.00005	0.0263
Q13564	*ULA1*	NEDD8-activating enzyme E1 regulatory subunit	−2.566	0.169	0.00004	0.0263
P07900	*HS90A*	Heat shock protein HSP 90-alpha	−1.587	0.333	0.00004	0.0263
Q9BQA1	*MEP50*	Methylosome protein 50	−2.31	0.201	0.00007	0.0275
P83731	*RL24*	60S ribosomal protein L24	−1.928	0.263	0.00010	0.0334
P46781	*RS9*	40S ribosomal protein S9	−1.413	0.375	0.00012	0.0334
Q6P9B6	*MEAK7*	MTOR-associated protein MEAK7	1.410	2.657	0.00016	0.0344
P27348	*1433T*	14-3-3 protein theta	−2.040	0.243	0.00051	0.0764
P28482	*MK01*	Mitogen-activated protein kinase 1	−1.636	0.322	0.00089	0.0809
P08238	*HSP90B*	Heat shock protein HSP 90-β (HSP 90B)	−1.601	0.330	0.00070	0.0809
P09525	*ANXA4*	Annexin A4	0.612	1.529	0.00101	0.0837

**Table 4 cells-12-02524-t004:** List of different categories of differentially expressed selectively enriched proteins in DM vs. N LEC-derived Exos with their gene and protein names, fold changes (FC), and *p* values.

Category	UniprotID	Gene	Protein Functional Name	FC	*p* Value
Endosomal Sorting Complexes Required for Transport	Q9UN37	*VPS4A*	Vacuolar protein sorting-associated protein 4A	0.3276	0.0018
	Q9UBQ0	*VPS29*	Vacuolar protein sorting-associated protein 29	0.2920	0.0022
Cargo selection	Q13564	*ULA1*	NEDD8-activating enzyme E1 regulatory subunit.	0.1689	0.0001
Trafficking / sorting	Q9H0U4	*RAB1B*	Ras-related protein Rab-1B	0.3474	0.0047
	P61106	*RAB14*	Ras-related protein Rab-14	0.4007	0.0055
	Q92930	*RAB8B*	Ras-related protein Rab-8B	2.2922	0.0133
	P61019	*RAB2A*	Ras-related protein Rab-2A	0.4009	0.0162
	Q13637	*RAB32*	Ras-related protein Rab-32	2.1274	0.0416
Heat shock proteins	P34932	*HSP74*	Heat shock 70 kDa protein 4	0.3953	0.0012
	P08238	*HS90B*	Heat shock protein HSP 90-beta	0.3296	0.0007
	O95757	*HS74L*	Heat shock 70 kDa protein 4L	0.3776	0.0021
	P07900	*HS90A*	Heat shock protein HSP 90-alpha	0.3329	0.00004
Chaperones	P27348	*1433T*	14-3-3 protein theta	0.2432	0.0005
Mitochondrial proteins	P06576	*ATPB*	ATP synthase subunit beta, mitochondrial	0.3929	0.0202
	O00159	*MYO1C*	Unconventional myosin-Ic	1.8061	0.0227
Integral membraneproteins	P29317	*EPHA2*	P29317	2.5968	0.0251
RNA binding proteins	P12429	*ANXA3*	Annexin A3	2.4456	0.0085
	P09525	*ANXA4*	Annexin A4	1.5290	0.0010
	P07355	*ANXA2*	Annexin A2	1.7087	0.0015

## Data Availability

The exosome genomics RNA-seq dataset is available from the public GEO repository under accession No. GSE243345. (https://www.ncbi.nlm.nih.gov/geo /query/acc. cgi?acc=GSE243345) accessed on 19 October 2023. The corneal limbus genomics small RNA-seq dataset is available from the public GEO repository under accession No. GSE97069,(https://0-www-ncbi-nlm-nih-gov.brum.beds.ac.uk/geo/query/acc.cgi?acc=GSE97069) accessed on 17 October 2023. The mass spectrometry proteomics data have been deposited to the ProteomeXchange Consortium via the PRIDE [86] partner repository with the dataset identifier PXD040918 (http://www.ebi.ac.uk/pride/archive/projects/PXD040918) accessed on 17 October 2023.

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
