# Peer review of "MicroRNA and Protein Cargos of Human Limbal Epithelial Cell-Derived Exosomes and Their Regulatory Roles in Limbal Stromal Cells of Diabetic and Non-Diabetic Corneas"

_cells, 2023, doi:10.3390/cells12212524_

Round 1

Reviewer 1 Report

This is a highly novel study which provides new information on the differences between exos from diabetic and nondiabetic limbal SCs. The data is solid, the comments are mainly aimed at filing in some missing details and improving the data presentation.

Methods:

-         - Have the authors also measured and compared the expression of keratocyte markers ALDH3A1 and lumican, MSC markers CD73, CD90 and CD105 in DM-LSC? OK, if not, just good to mention one way or another.

-          Were EVs were collected every other day? How many times was this repeated for each flask? Did you pool the EVs at the end from one flask?

Results:

Results 3.1.

-          Authors showed characterization of primary limbal stromal cells only in non-diabetic human primary in supplementary S1B. Was this also done for diabetic primary cells. Would be good to comment on any differences.

Result 3.2.

-          The paragraph on the results of NanoSight technology, flow cytometry, and western blot, could be improved by including more specific quantifications, such as the actual particle sizes measured by NanoSight, the quantification of CD63 and CD81 expression, and the intensity of bands in the western blot.

-          The paragraph mentions "no significant difference" between non-diabetic and diabetic LEC-derived Exos, it would be good to include statistical information (e.g., p-values). The paragraph states that "the majority of our isolated EVs were exosomes" but does not explain the significance of this finding in the context of the study. Providing some context or interpretation of the results would help the reader understand their importance.

-          It is not easy to follow which experiments were conducted regarding markers. Please clarify which marker were only used for flow cytometry and which ones for western blot. Did they perform CD81 western blot? It would be good to provide the percentage of expression of each marker by flow cytometry data to demonstrate the purity of the cell population.

-          The western blot data for HSP70 is not shown in fig 2 although it is mentioned in the results section 3.2. if authors have data for each specific marker for both flow and western, it would be good to show them, otherwise OK ­

Result 3.3.

-          The paragraph is somewhat long and could benefit from greater conciseness to help the reader follow the flow of the experiment and the results presented.

-          Figure 2 C and D would be clearer with labelling. Immunostaining of LEC-derived exosome (Exo) uptake by non-diabetic (N) and diabetic (DM) LSCs is shown but Immunostaining of LSC-derived exosome (Exo) uptake by non-diabetic (N) and diabetic (DM) LSCs is not shown (would be good to show if available). The paragraph mentions a "five-fold higher internalization" of LEC-derived Exos compared to LSC-derived Exos, but it doesn't provide the actual quantitative data (e.g., mean fluorescence intensity values or percentages).

-   -          Authors have mentioned that they have examined LSC-LEC crosstalk by Exos, we showed the uptake of exogenous PKH-labeled LEC- or LSC-derived Exos by primary LSC. Did they also examine the uptake of LEC or LSC derived Exos by LECs?

-          Authors mentioned that they have found The proliferation rate was also significantly increased in N- and DM-Exos treated primary N-LSCs compared to untreated cells, whereas there were no significant changes in DM-LSC proliferation rate treated with DM-LEC-derived Exos compared to untreated cells, but I cant seem to find the data. 

-          Is number 5 before conclusion a typo?

Some editing to shorten some of the longer paragraphs would be helpful

Reviewer 2 Report

This is a very interesting article 

Author Response

This is a very interesting article.

Thank you!

Reviewer 3 Report

Dr. Verma et., al. authors have investigated whether the exosomes play a role in regulating corneal pathogenesis during diabetes. They have differentially and extensively characterized the contents of miRNA and proteins of human limbal epithelial cell-derived exosomes. They also tested their potential biological function in vitro in LSC cellular proliferation and wound the scratch wound assays. The studies are well designed, and the results are very interesting, which may provide the molecular base for diabetic exosome therapy. 

Minor concern:

1.        The images of Figure 2A-C, Figure 3-4, and Figure 7 are not presented well, or the fonts and images are too small to see. The original tiff files may be used here without loss of their resolution.

Author Response

The images of Figure 2A-C, Figure 3-4, and Figure 7 are not presented well, or the fonts and images are too small to see. The original tiff files may be used here without loss of their resolution

Thank you for your suggestion, we are now providing the figures with higher resolutions

Round 2

Reviewer 1 Report

The authors have addressed all the comments very sell